# Lactate Rewrites the Metabolic Reprogramming of Uveal Melanoma Cells and Induces Quiescence Phenotype

**DOI:** 10.3390/ijms24010024

**Published:** 2022-12-20

**Authors:** Lucia Longhitano, Sebastiano Giallongo, Laura Orlando, Giuseppe Broggi, Antonio Longo, Andrea Russo, Rosario Caltabiano, Cesarina Giallongo, Ignazio Barbagallo, Michelino Di Rosa, Rosario Giuffrida, Rosalba Parenti, Giovanni Li Volti, Nunzio Vicario, Daniele Tibullo

**Affiliations:** 1Department of Biomedical and Biotechnological Sciences, University of Catania, 95123 Catania, Italy; 2Department of Medical, Surgical Sciences and Advanced Technologies “G.F. Ingrassia”, Anatomic Pathology, University of Catania, 95123 Catania, Italy; 3Department of Ophthalmology, University of Catania, 95123 Catania, Italy

**Keywords:** uveal melanoma, lactate, lactylation, HCAR1, metabolism

## Abstract

Uveal melanoma (UM), the most common primary intraocular cancer in adults, is among the tumors with poorer prognosis. Recently, the role of the oncometabolite lactate has become attractive due to its role as hydroxycarboxylic acid receptor 1 (HCAR1) activator, as an epigenetic modulator inducing lysine residues lactylation and, of course, as a glycolysis end-product, bridging the gap between glycolysis and oxidative phosphorylation. The aim of the present study was to dissect in UM cell line (92.1) the role of lactate as either a metabolite or a signaling molecule, using the known modulators of HCAR1 and of lactate transporters. Our results show that lactate (20 mM) resulted in a significant decrease in cell proliferation and migration, acting and switching cell metabolism toward oxidative phosphorylation. These results were coupled with increased euchromatin content and quiescence in UM cells. We further showed, in a clinical setting, that an increase in lactate transporters MCT4 and HCAR1 is associated with a spindle-shape histological type in UM. In conclusion, our results suggest that lactate metabolism may serve as a prognostic marker of UM progression and may be exploited as a potential therapeutic target.

## 1. Introduction

Uveal melanoma (UM) has been classified as a rare disease, but it is still the most common intraocular cancer in adults, with about 7095 new cases per year worldwide [1,2,3,4]. UM and cutaneous melanoma (CM) are both characterized by an aberrant growth of melanocytes, even if UM retains a typical biological and genetic signature. While CM is often characterized by mutations on *BRAF*, *NRAS*, *or KIT*, UM patients usually carry the mutated GPCR alpha subunits *GNAQ* or *GNA11*. Further evaluations reported the inactivating somatic mutations in the gene encoding for BRCA-1-associated protein 1 (BAP1) [5,6,7,8,9]. As a result, 84% of BAP1-mutated patients are prompted to develop liver (89%), lung (29%), and bone (17%) cancer, with a prognosis of ~15% upon 5 years [7,10]. Therefore, the current estimation is that 40–50% of UM patients will die of metastatic disease, even with early diagnosis and proper treatment [11]. A number of clinical, histopathological, and cytogenetic features have been reported to be valuable prognostic factors predicting UM progression [12,13,14,15]. However, there is still a lack of proper treatments aiming at counteracting tumor progression [16]. For this purpose, previous studies reported programmed cell death as an outstanding factor related to tumorigenesis, progression, and metastasis processes [17,18,19]. In addition, the tumor microenvironment (TME) turned out to also be a key player in such processes. Indeed, the milieu in which tumors are located contains numerous non-tumor cell types, such as immune cells, inflammatory cells, mesenchymal cells, and endothelial cells, exerting physiological functions, but eventually acting as pro-tumoral players [20,21,22,23]. It is worth noting that bystander cell populations also act by reshaping the phenotype of the malignant cells without altering their genetic signatures [24]. In this context, cancer cells exhibited an increased glycolytic rate, the so-called Warburg effect, resulting in a strong lactate production, in turn serving as an oncometabolite prompting tumor progression and metastasis [25,26], while suppressing both innate and adaptive immune cell response [27,28]. In this regard, lactate has been reported to act via hydroxycarboxylic acid receptor 1 (HCAR1) in synergy with monocarboxylate transporters 1–4 (MCT 1–4). Corroborating the role of lactate as an oncometabolite, several studies reported HCAR1 targeting as one of the main factors leading to pancreatic and breast cancer progression [29,30,31]. Further studies also uncovered an epigenetic trait covered by lactate, which may modify lysine residues by lactylation, or it may modulate histone deacetylases (HDACs) and histone acetyl transferases (HATs) activity [32,33,34,35]. As a result, lysine lactylation or acetylation disrupts the electrostatic interaction standing between histones and DNA, triggering a permissive chromatin, eventually promoting DNA damage repair (DDR) [36,37,38,39]. This body of evidence points to lactate metabolism modulation as a potential strategy toward the development of efficient drugs leading the path against tumor [40].

In this study, we aimed at investigating the accumulation of lactate within UM TME, thus dissecting its role as a oncometabolite and eventually uncovering new targets toward the development of more effective UM drugs.

## 2. Results

### 2.1. Lactate and HCAR1 Targeting Exerts Opposite Effects in Uveal Melanoma Cell Line

In order to assess how lactate accumulation affects UM progression, we supplemented lactate 20 mM on a 92.1 UM cell line model. We observed a significant decrease in the normalized cell index after lactate exposure confirmed by a decrease in the total AUC compared to untreated cells (Figure 1A). We then compared the effect of increased levels of extracellular lactate with the selective stimulation of the lactate receptor GPR81 (HCAR1) mediated by 3,5-DHBA, at the final concentration of 150 μM. Interestingly, we detected that a selective stimulation of the HCAR1 receptor produced an opposite effect compared to lactate treatment, resulting in an increase in the normalized cell index compared both to lactate and untreated cells, confirmed also by an increase in the total AUC (Figure 1A). Subsequently, we analyzed the effect of lactate and receptor stimulation on cell migration. Our results show a significant increase in the percentage of wideness in the scratch assay at 24 and 48 h in lactate-treated cells (Figure 1B,C). HCAR1 stimulation, on the other hand, significantly decreased the percentage of wideness in the scratch assay at 24 and 48 h in treated cells as compared to untreated and lactate-treated cells (Figure 1B,C). Overall, these data indicate a prominent role of lactate as a metabolite in inhibiting cell proliferation rather than as a signaling molecule acting on HCAR1.

### 2.2. Inhibition of Lactate Uptake Induces Uveal Melanoma Growth

In order to further investigate how lactate affects UM progression, we analyzed the effect of the MCT1 inhibitor (AZD3965, 10 mM) and the HCAR1 antagonist (3-OBA, 3 mM) on 92.1 UM cell proliferation and migration.

Our results show that treatments with both AZD3965 and 3-OBA had no effect on cell proliferation, as indicated by the normalized cell index and AUC values compared to untreated control cells (Figure 2A–D). On the one hand, cultures cotreated with lactate and AZD3965 resulted in an increased normalized cell index value and AUC value as compared to untreated control cells and to lactate or AZD3965 single treatment (Figure 2A). On the other hand, cultures exposed to lactate and 3-OBA in cotreatment showed a decreased normalized cell index and AUC value as compared to untreated control cells and significantly increased as compared to lactate single treatment (Figure 2D).

These results were confirmed by wound-healing assay (Figure 2B,C,E,F), showing that the cotreatment with lactate and AZD3965 resulted in a significant decrease in the percentage of wideness as compared to lactate alone (Figure 2B,C), as well as the cotreatment with lactate and 3-OBA resulted in a significant decrease in the percentage of wideness as compared to lactate-treated cells (Figure 2E,F).

### 2.3. Lactate Treatment Increases HCAR1 and Lactate Transporters in Uveal Melanoma

We previously reported that lactate supplementation affects the expression of its transporter proteins [41]. Corroborating these data in a different in vitro model, our results show that lactate treatment was able to induce a significant increase in mRNA expression levels of *SLC16A1* (gene encoding MCT1) and *HCAR1*, and that these effects were reverted by AZD3965 (Figure 3A,B). These data were further confirmed by Western blot analysis, showing an increase in the protein expression levels of MCT1 and HCAR1 in lactate-treated cells and a reduction in their expression in lactate and AZD3965 cotreated cultures (Figure 3C–E). Given the effect of lactate as a metabolite, on the expression of MCT1 and HCAR1, we subsequently analyzed its effect as a signal molecule through receptor inhibition. Interestingly, our data showed that a lactate and 3-OBA cotreatment resulted in a significant reduction in HCAR1 protein expression as compared to lactate-treated cells (Figure 3H–J). Taken together, these results support the hypothesis that lactate may prompt its own import, promoting the expression of lactate transporters.

### 2.4. Lactate Rewires Uveal Melanoma Metabolism Increasing mRNA Levels of Genes Involved in Mitochondrial Metabolism

Since lactate supplementation inhibits cellular growth, we thought to investigate whether this effect may be related to changes in cell metabolism. For this purpose, we analyzed a panel of mRNAs of genes involved in mitochondrial activity and energy metabolism. Our results show that lactate treatment, both upon 24 and 48 h, increased by about four-fold the relative mRNA levels of *PPARG coactivator 1 alpha* (*PGC1a*), *sirtuin 1* (*SIRT1*), *and transcription factor A*, *mitochondrial* (*TFAM*), associated with an overall increase in *ATP synthase* (*ATP syn*), *cytochrome c oxidase subunit 4* (*COX IV*), *COX II*, and *mitochondrial NADH-ubiquinone oxidoreductase chain 4* (*ND4*) (Figure 4A). Interestingly, we also observed a significant increase in mRNA expression levels of lactate dehydrogenase (*LDHA*) and *MCT4* in lactate-treated cells compared to untreated cells (Figure 4B). Given the effect on cell migration and proliferation of HCAR1 receptor stimulation, we analyzed the mRNA expression of the same genes even after treatment with the 3,5-DHBA agonist. Our analysis revealed that HCAR1 activation induced a significant increase in mRNA expression levels of *SIRT1*, *PGC1a*, *TFAM*, *ATPsyn*, *COXII*, *COX IV*, *ND4*, *and LDHA* (Figure 4C,D), but produced an opposite effect on *MCT4* expression compared to lactate (Figure 4D), suggesting that the activation of the HCAR1 receptor causes the metabolic switch of the UM cells toward oxidative metabolism.

Finally, we analyzed the genes involved in mitochondrial metabolism after receptor blockade via the 3-OBA antagonist (Figure 4E,F) to link HCAR1 stimulation with the effects on mitochondria observed in UM cell lines. Interestingly, our results show that the lactate and 3-OBA co-treatment reverts the lactate-mediated effect, eventually decreasing *LDHA* and MCT4 mRNA levels, along with *SIRT1*, *PGC1a*, *TFAM*, *COX II*, *COX IV*, and *ND4* (Figure 4E,F).

### 2.5. Lactate Supplementation Increases Euchromatin Rate and Quiescence in Uveal Melanoma Cells

Since lactate supplementation seems to impair UM progression, we thought to dissect a possible mechanism by which lactate may exert its effect. For this purpose, we analyzed by Operetta the chromatin relaxation index of 92.1 UM cells supplemented by lactate (Figure 5A). Quantification displayed a significant decrease in percentage of heterochromatin upon lactate supplementation (Figure 5B). Moreover, nuclei characterized by a relaxed chromatin also showed enlarged nuclei (Figure 5C). To further corroborate the role of lactate in promoting an euchromatic state, we thought to assess the level of histone lactylation in our sample. We found a significant increase in the H3K18lac level in UM cells treated with lactate as compared to untreated control cells (Figure 5D). An increased euchromatic rate along with a decreased cell growth and enlarged nuclei are three of the fingerprints associated with cellular quiescence. To corroborate this hypothesis, we tested the expression, by qPCR, of five quiescence markers, namely *p53*, *p21*, *CYTC*, *FOXO3*, and *EZH2*, which were all overexpressed upon lactate supplementation (Figure 5E). Overall, our results provide a mechanism behind lactate supplementation and cell growth blockage.

### 2.6. Patients Characterized by Spindle-Shape Histological Type Shows Increased MCT4 and HCAR1 Accumulation

To further corroborate our data indicating lactate as a metabolite able to inhibit cell proliferation, we assessed MCT4 and HCAR1 expression levels on epithelioid cells versus spindle cells of UM patients. Our results show a stronger signal in MCT4 and HCAR1 in spindle cells as compared to epithelioid UM sections, showing diffuse and weak staining for MCT4 and virtually absent HCAR1 (Figure 6). This evidence correlates with a less aggressive tumor phenotype.

## 3. Discussion

Despite its classification as a rare disease, UM is still the most common adult intraocular cancer. To date, valuable prognostic markers and a proper treatment aiming at counteracting its progression are still missing [42,43]. For this purpose, understanding the molecular mechanisms underlying UM progression may unveil new factors potentially serving as prognostic or targetable factors. Our previous reports showed how macroH2A1, an epigenetic factor involved in cell differentiation and the establishment of a heterochromatic state, may be an outstanding factor marking UM progression [44,45]. Furthermore, it is well established that cancer may rewire its metabolism in order to promote its progression and cell fate [46,47]. Such a phenomenon is strongly affected by intercellular communication, exchanges, and milieu conditioning of bystander cell populations, which may significantly interfere with differentiation and regenerative processes, sustaining tumorigenesis and cell invasion [48,49,50,51,52,53,54]. Indeed, the milieu in which tumors are located contains numerous non-tumor cell types, such as immune cells, inflammatory cells, mesenchymal cells, and endothelial cells, exerting physiological functions, but eventually acting as pro-tumoral players [20,21,55].

In this work, we provided new insights on the role of lactate as a metabolite potentially regulating UM progression. In this regard, we previously reported that lactate accumulation may drive glioblastoma progression, serving as an oncometabolite promoting tumor progression [41]. Interestingly, our results show that in UM, lactate supplementation impairs tumor growth. In particular, evidence have been provided on how lactate behaves either as signaling molecule by HCAR1-mediated cascade, and as a proper metabolite through its import and export channels, namely MCT1 and MCT4, respectively [41,56]. Our results depict a scenario in which lactate impairs the growth of UM acting via MCT1, rather than modulating HCAR1 cascade. These data are further supported by our results, showing a strong increase in UM cell growth upon treatment with MCT1 selective inhibitor AZD3965, eventually impairing lactate uptake. Interestingly, ours and other groups already investigated the role of MCT1 in predicting tumor progression, using different cancer models [41,57,58,59]. However, while in these systems a positive correlation between MCT1 expression and cancer progression has been reported, our data on UM cells display the opposite trend, possibly relating to the role played by MCT1 on the tumor context. Furthermore, MCT1 and MCT4 expression are strictly related to an increased lactate level [41,60,61,62,63]. Interestingly, these data are supported by recent evidence showing that lactate supplementation boosts the expression of its transporters, along with a boost in OXPHOS activity [64,65]. Corroborating these data, our RT-qPCR results also show an increase in OXPHOS-related genes. Of note, cancer cells relying on an OXPHOS-based metabolism usually decrease their metastatic potential, as also supported by our scratch assay analysis and recently reviewed [66]. These data are also corroborated by our ex vivo analysis on UM tissue. Here, we showed that epithelioid cells, associated with a better prognosis, display enhanced MCT4 accumulation compared to the more aggressive spindle-cell specimens, as also supported by our GEO dataset analysis [67]. To dissect the mechanism by which lactate decreases cell proliferation, we show here for the first time that its supplementation enhances H3K18 lactylation on UM cells, a phenomenon by which lactate promotes an euchromatin state, also characterized by enlarged nuclei. Therefore, it has been reported that quiescent cells display a reduced proliferation, a more relaxed chromatin, and they may rely on oxidative phosphorylation over glycolysis [68,69]. Conversely, our data showed an increased expression of quiescent-related markers, corroborating our model and depicting a scenario in which lactate impairs UM growth, prompting cellular quiescence.

Overall, this work highlighted how lactate plays a role strongly dependent on the tumor context (Figure 7). In the UM cellular model, we showed that increase in MCT4 and HCAR1 expressions were strictly related to the spindle-shaped histological type.

## 4. Materials and Methods

### 4.1. Cell Culture and Pharmacological Treatments

Human uveal melanoma cells (92.1) were purchased from ATCC Company (Milan, Italy). Briefly, cells were cultured in RPMI1640 medium with 10% fetal bovine serum (FBS) (cat. no. 10082147), 100 U/mL penicillin, and 100 U/mL streptomycin (cat. no. 15070063; all from Gibco, Waltham, MA, USA) and expanded once they reached 80% confluency using trypsin-EDTA solution (0.05% trypsin and 0.02% EDTA). Lactate (Sigma-Aldrich, Milan, Italy), AZD3965, 3,5-dihydroxybenzoic acid (3-5-DHBA) (Sigma-Aldrich, Milan, Italy), and 3-hydroxybutyric acid (3-OBA) (Sigma-Aldrich, Milan, Italy) were added to cell culture at final concentrations of 20 mM, 10 μM, 150 μM, and 3 mM, respectively, when needed.

### 4.2. RNA Extraction and RT-qPCR

Total RNA extraction was performed as previously described [70], using Trizol^®^ reagent (category no. 15596026, Invitrogen, Carlsbad, CA, USA). cDNA was synthesized by High-Capacity cDNA Reverse Transcription kit (category no. 4368814, Applied Biosystems, Foster City, CA, USA). RT-qPCR was performed using Step-One Fast Real-Time PCR (Applied Biosystems, Foster City, CA) and SYBR Green PCR MasterMix (category no. 4309155, Life Technologies, Monza, Italy).(primers’ sequences are shown in Table 1). Primers (Table 1) were purchased from Metabion International AG (Planneg, Germany).

### 4.3. Western Blot Analysis

Protein detection was performed by incubating MCT1 (1:1000; AB90582, Abcam, Cambridge, UK) and β-actin (1:1000; anti-mouse, cat. no. 4967S; Cell Signalling Technology, Milan, Italy) overnight at 4 °C. For histone protein extraction, we used Abcam histone extraction kit (AB113476, Abcam, Cambridge, UK) according to manufacturer’s protocol. For protein detection, rabbit primary H3K18Lac (1:1000; PTM-1406, PTM-biolabs, IL, USA) and H3 (1:1000; AB18521, Abcam, Cambridge, UK) were used. The next day, the membranes were washed three times in PBS for 5 min and incubated with secondary infrared anti-mouse IRDye800CW (1:5000) in PBS/0.5% Tween-20 for 1 h at room temperature. All antibodies were diluted in Odyssey Blocking Buffer. Protein bands intensity was quantified and normalized to β-actin levels [71,72,73,74].

### 4.4. Real-Time Monitoring of Cell Proliferation

xCELLigence experiments were performed using the Real-Time Cell Analysis (RTCA) dual plate (DP) instrument (Roche Applied Science, Mannheim, Germany, and ACEA Biosciences, San Diego, CA, USA) as previously described [75]. Briefly, the optimal seeding number was determined by cell titration and growth experiments. After seeding the optimal cell number (3000 cells/well), the cells were treated with lactate, AZD3965, 3,5-DHBA, and 3-OBA, and automatically monitored every 15 min for 24 h.

### 4.5. Effects of Pharmacological Treatments on Cell Migration

Cell migration was examined by employing the wound-healing assay [76]. Briefly, cells were seeded in 24-well dishes and cultured until confluence. At this stage, lactate, AZD3965, 3,5-DHBA, or 3-OBA were added where needed and cell culture was scraped with a 200 mL micropipette tip. Wound closure was detected at 0, 24, and 48 h. The uncovered wound area was measured and quantified at different intervals with ImageJ v1.37 (NIH, Bethesda, MD, USA).

### 4.6. Immunocytochemistry Analysis

Immunocytochemistry was carried out as previously reported [77]. Briefly, mitochondria were stained with 200 nM MitoTracker Red CMXRos probe (Thermo Fisher Scientific, Milan, Italy) for 30 min at 37 °C, according to the manufacturer’s instructions. Cells were treated with the dye for 30 min at 37 °C, and it was removed after 30 min. At this stage, cells were washed 3 times in phosphate-buffered saline (PBS) to remove the unbound probe. Nuclei were stained by NucBlue (two drops per mL) (Thermo Fisher Scientific, Milan, Italy) for 15 min at 37 °C, according to the manufacturer’s instructions. Finally, cells were treated with lactate 20 mM. For image acquisition, we used Operetta (Perkinelmer, MA, USA), where cells were maintained at 37 °C and images were captured at 24 h after treatment. Data collected were analyzed by Harmony software (Perkinelmer, MA, USA).

### 4.7. Patients’ Cohort

Primary UM samples were retrospectively collected after they were surgically enucleated from October 2009 to October 2019 at the Ophthalmologic Clinic of the University of Catania. For all of them, enucleation was the only treatment option. As previously described [67], the corresponding clinical pathological data were retrieved from the original pathological reports. The present research complied with the Helsinki Declaration and all experiments were approved by the local Ethics Committee, Comitato Etico Catania 1, University of Catania (ID: 003186-24). The previously reported criteria of exclusion were used for case selection [78].

### 4.8. Immunohistochemical Analysis

Immunohistochemical analysis was performed as previously described [79]. Briefly, deparaffinized and pretreated slides were incubated for 30 min at 37 °C with MCT4 and HCAR1 (1:1000; AB90582, Abcam, Cambridge, UK) antibody. Immunostaining specificity was assayed omitting antibodies.

### 4.9. Statistical Analysis

Statistical analysis was performed using Prism Software using Mann–Whitney U test for comparison of *n* = 2 groups. For comparison of *n* ≥ 3 groups, one-way or two-way analysis of variance (ANOVA) with Holm–Sidak post hoc test for multiple comparisons were used where appropriate (Graphpad Software Inc., California, USA, RRID: rid_000081). Data are expressed as mean ± SD, unless otherwise stated. For all statistical tests, *p*-values < 0.05 were considered statistically significant.

## Figures and Tables

**Figure 1 ijms-24-00024-f001:**
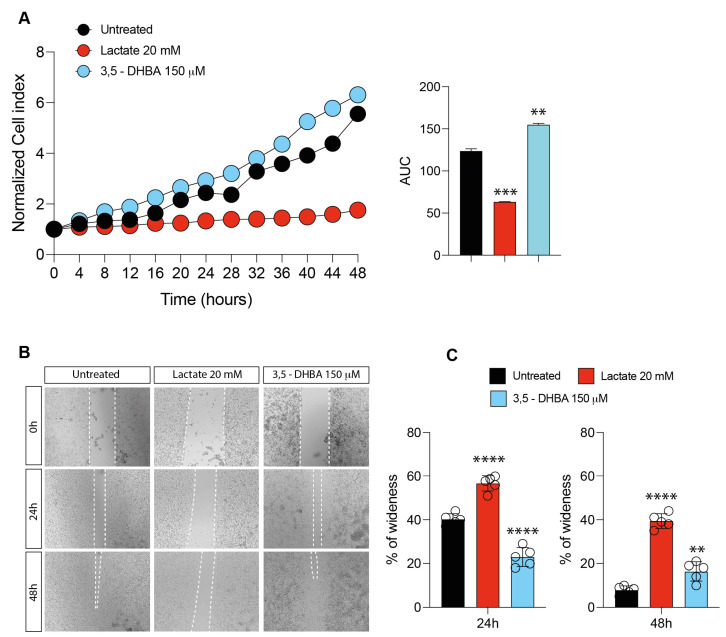
Effect of lactate and 3,5-dihydroxybenzoic acid (3,5-DHBA) on uveal melanoma cell proliferation and migration. (**A**) Real-time cell proliferation monitoring in 92.1 cells, using the xCELLigence system following treatments with lactate (20 mM) and 3,5-DHBA (150 μM). The cell index values were normalized at the time of pharmacological treatments in order to obtain a normalized cell index. Each dot expresses the average of four different experiments and the area under curve (AUC) is also reported. (**B**,**C**) Representative micrograph (**B**) and quantification (**C**) of human uveal melanoma cell migration analysis with the wound-healing assay following treatments with lactate (20 mM) and 3,5-DHBA (150 mM). Data are mean of five independent experiments ± SD (one-way ANOVA). ** *p* < 0.01; *** *p* < 0.001; **** *p* < 0.0001.

**Figure 2 ijms-24-00024-f002:**
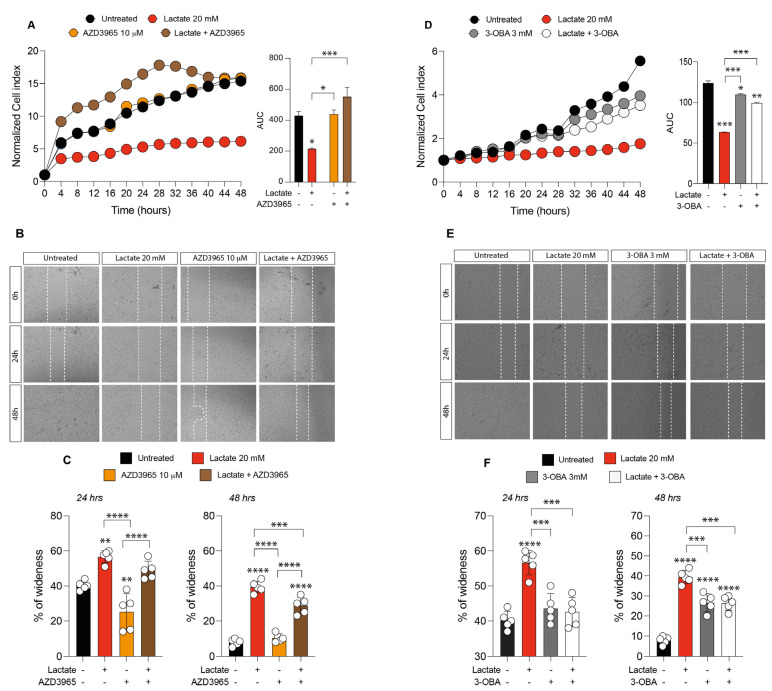
Effect of AZD3965 and 3-hydroxy-butyrate acid (3-OBA) on uveal melanoma cell proliferation and migration. (**A**) Real-time cell proliferation monitoring in 92.1 cells, using the xCELLigence system following treatments with lactate (20 mM) and AZD3965 (10 μM); the cell index values were normalized at the time of pharmacological treatments in order to obtain a normalized cell index. Each dot expresses the average of four different experiments. (**B**,**C**) Representative micrograph (**B**) and quantification (**C**) of human uveal melanoma cell migration analysis with the wound-healing assay following treatments with lactate (20 mM) and AZD3965 (10 μM). Data are mean of three independent experiments ± SD. (**D**) Real-time cell proliferation monitoring in 92.1 cells, using the xCELLigence system following treatments with lactate (20 mM) and 3-OBA (3 mM); the cell index values were normalized at the time of pharmacological treatments in order to obtain a normalized cell index. Each dot expresses the average of four different experiments. (**E**,**F**) Representative micrograph (**E**) and quantification (**F**) of human uveal melanoma cell migration analysis with the wound-healing assay following treatments with lactate (20 mM) and 3-OBA (3 mM). Data are mean of five independent experiments ± SD (two-way ANOVA). * *p* < 0.05; ** *p* < 0.01; *** *p* < 0.001; **** *p* < 0.0001.

**Figure 3 ijms-24-00024-f003:**
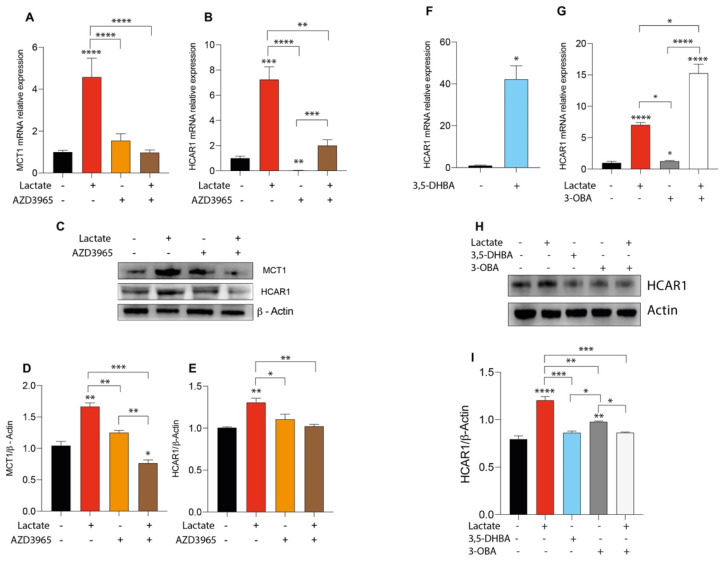
Effect of lactate and 3,5-dihydroxybenzoic acid (3,5-DHBA), AZD3965 and 3-hydroxy-butyrate acid (3-OBA) in expression of monocarboxylate transporter 1 (MCT1) and HCAR1 in uveal melanoma cell line. (**A**,**B**) *MCT1* and *HCAR1* mRNA expression levels following 24 h of lactate (20 mM) and AZD3965 (10 μM) treatment. (**C**–**E**) MCT1 and HCAR1 protein expression levels following 24 h of lactate (20 mM) and AZD3965 (10 μM) treatment. (**F**,**G**) HCAR1 mRNA expression levels following 24 h of 3,5-DHBA (150 μM), lactate (20 mM), and 3-OBA (3 mM) treatment. (**H**,**I**) HCAR1 protein expression levels following 24 h of 3,5-DHBA (150 μM), lactate (20 mM), and 3-OBA (3 mM) treatment. Values represent the mean ± SD of experiments performed in quadruplicate. The figures presented are representative of four independent experiments, and values represent the mean ± SD of experiments performed in quadruplicate (Mann–Whitney U test or two-way ANOVA). * *p* < 0.05; ** *p* < 0.01; *** *p* < 0.001; **** *p* < 0.0001.

**Figure 4 ijms-24-00024-f004:**
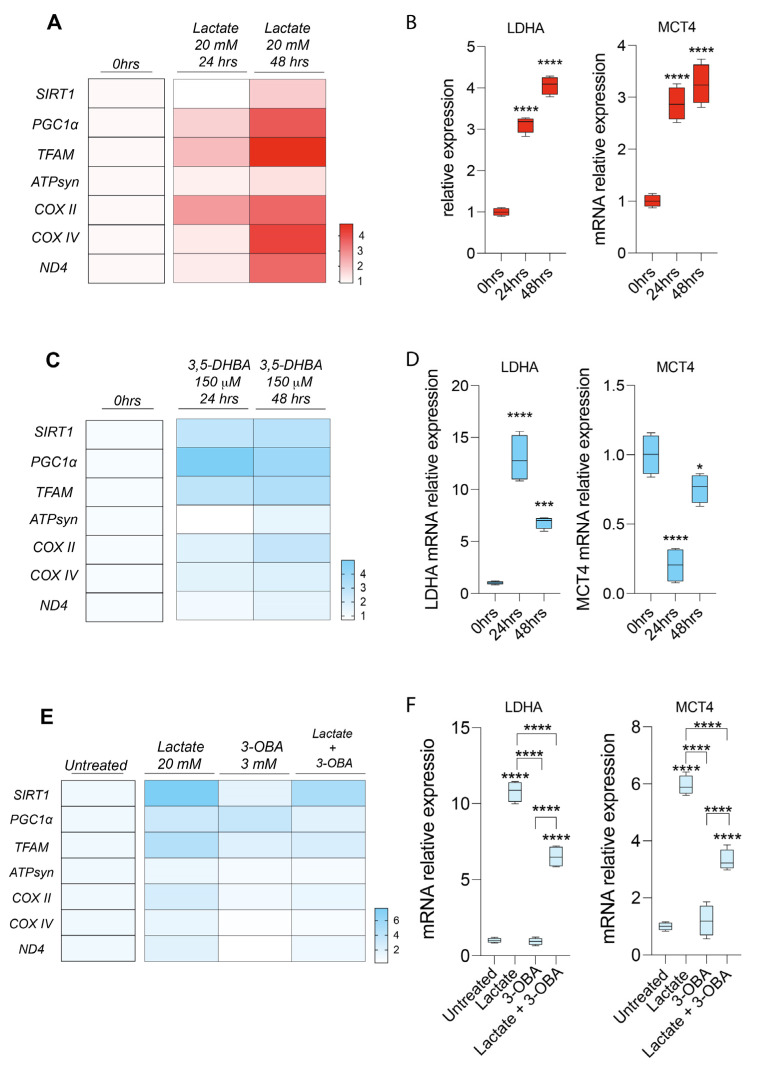
Effect of lactate, 3,5-dihydroxybenzoic acid (3,5-DHBA), and 3-hydroxy-butyrate acid (3-OBA) in the expression of genes involved in mitochondrial metabolism. (**A**,**B**) mRNA expression levels of *SIRT1*, *PGC1a*, *TFAM*, *ATPsyn*, *COX II*, *COXIV*, *ND4*, *LDHA*, and *MCT4* following 24 h and 48 h of lactate (20 mM) treatment. (**C**,**D**) mRNA expression levels of *SIRT1*, *PGC1a*, *TFAM*, *ATPsyn*, *COX II*, *COXIV*, *ND4*, *LDHA*, and *MCT4* following 24 h and 48 h of 3,5-DHBA (150 μM) treatment (one-way ANOVA). (**E**,**F**) mRNA expression levels of *SIRT1*, *PGC1a*, *TFAM*, *ATPsyn*, *COX II*, *COXIV*, *ND4 LDHA*, and *MCT4* following 24 h of lactate (20 mM) and 3-OBA (3 mM) treatment. Data are shown via standard box-and-whiskers plot (two-way ANOVA). * *p* < 0.05; *** *p* < 0.001; **** *p* < 0.0001.

**Figure 5 ijms-24-00024-f005:**
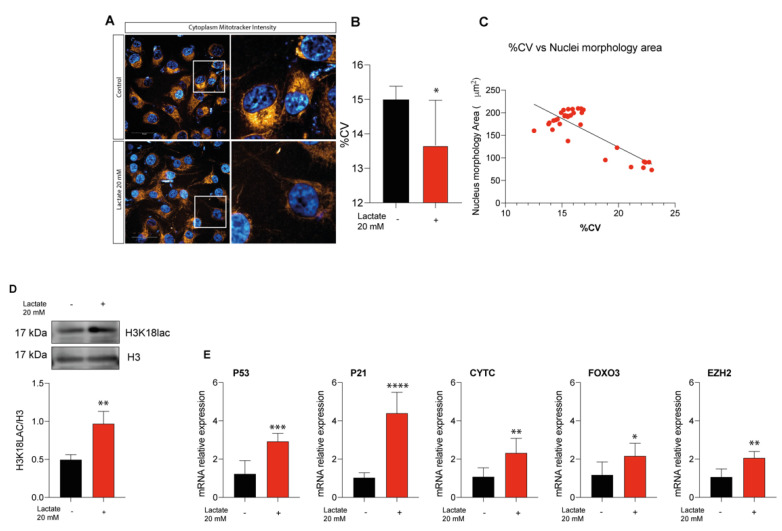
Lactate treatment reshaped 92.1 chromatin architecture. (**A**) Representative images showing Operetta analysis on UM cells untreated (top left) or under lactate (20 mM) treatment (bottom left). (**B**) Quantitative analysis of NucBlue coefficient of variance (CV) following lactate (20 mM) treatment. (**C**) Correlative analysis relating NucBlue %CV and Nuclear morphology area. (**D**) H3K18lac protein expression level following 24 h of lactate (20 mM) treatment. (**E**) mRNA expression levels of cellular senescence markers (*P53*, *P21*, *CYTC*, *FOXO3*, *and EZH2*) following 24 h of lactate (20 mM) treatment. Values represent the mean ± SD of experiments performed in quadruplicate. Data are representative of four independent experiments, and graphs are mean ± SD of experiments performed in quadruplicate (Mann–Whitney U test). * *p* < 0.05; ** *p* < 0.01; *** *p* < 0.001; **** *p* < 0.0001.

**Figure 6 ijms-24-00024-f006:**
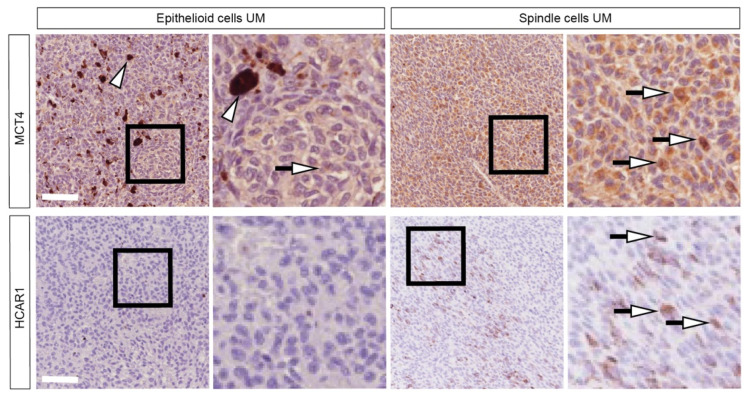
Lactate transporter MCT4 and lactate receptor HCAR1 are overexpressed in advanced UM. Representative pictures of the MCT4 and HCAR1 expression in human biopsies of patients with epithelioid and spindle UM. Arrowheads indicate dark-brown pigmentation and arrows indicate positive staining. Scale bars: 50 μm.

**Figure 7 ijms-24-00024-f007:**
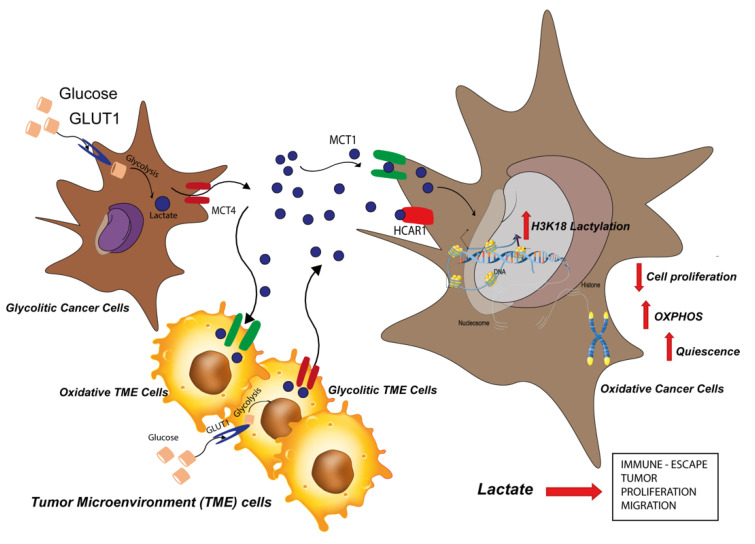
Lactate reshapes the metabolic reprogramming and induces quiescence phenotype in uveal melanoma cells.

**Table 1 ijms-24-00024-t001:** Primers’ list.

Primer	$\mathrm{Forward}\;(5^{\prime }\to 3^{\prime })$	$\mathrm{Reverse}\;(5^{\prime }\to 3^{\prime })$	Accession Number
PGC1alpha	ATGAAGGGTACTTTTCTGCCCC	GGTCTTCACCAACCAGAGCA	NM_001330751.2
SIRT1	AGGCCACGGATAGGTCCATA	GTGGAGGTATTGTTTCCGGC	NM_012238.5
COX IV	CAGCTCTCGGAAGCGTTGTA	GATAACGAGCGCGGTGAAAC	NM_001318802.2
SLC16A1	TGTTGTTGCAAATGGAGTGT	AAGTCGATAATTGATGCCCATGCCAA	NM_003051.4
SLC16A3	TATCCAGATCTACCTCACCAC	GGCCTGGCAAAGATGTCGATGA	NM_001206950.2
HCAR1	TTCGTATTTGGTGGCAGGCA	TTTCGAGGGGTCCAGGTACA	NM_032554.4
LDHA	GGATCTCCAACATGGCAGCCTT	AGACGGCTTTCTCCCTCTTGCT	NM_005566.4
ATP5F1A	CCGCCTTCCGCGGTATAATC	ATGTACGCGGGCAATACCAT	NM_001001937.2
P53	CTACAGTACTCCCCTGCCCT	GGGGCCAGACCATCGCTA	NM_001276697.3
P21	GTCAGTTCCTTGTGGAGCCG	GCCATTAGCGCATCACAGTC	NM_001374511.1
CYTC	CCGCCAATAAGAACAAAGGCATC	ATAAGGCAGTGGCCAATTATTACTC	NM_018947.6
FOXO3	GTGTTCCAGGGGAAGCACAT	GCTCTTGCCAGTTCCCTCAT	MK390615.1
EZH2	GACTGCTTCCTACATCGTAAGTG	CTTTGCTCCCTCCAAATGCT	XM_011515892.2
β-Actin	CCTTTGCCGATCCGCCG	AACATGATCTGGGTCATCTTCTCGC	NM_001101.5

## Data Availability

All data are included in the present manuscript.

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
