# Peer review of "Lactate Rewrites the Metabolic Reprogramming of Uveal Melanoma Cells and Induces Quiescence Phenotype"

_ijms, 2022, doi:10.3390/ijms24010024_

Round 1
Reviewer 1 Report
The present study aimed to dissect in UM cell line (92.1) the role of lactate as either a metabolite or a signalling molecule using known modulators of HCAR1 and of lactate transporters. The authors found that lactate resulted in a significant decrease of cell proliferation and migration acting and switching cell metabolism towards oxidative phosphorylation. They showed the results were coupled with increased euchromatin content and quiescence in UM cells.
1. In figure 3, the authors showed that lactate treatment induced a significant increase in mRNA expression and protein levels of MCT1 and HCAR1 and that these effects were reverted by AZD3965.Please explain how MCT1 the inhibitor block lactate induced the expression of MCT1. In Figures 3h and 3i, there is no big difference between the different groups.
2. In figure 4, how about the expression of ND4, TFAM.
3. Why did the authors choose to detect the effect of lactate on the expression H3K18lac? And in figure 5, I suggest the authors detect the senescence markers in a different way to confirm the results.
4. In Figure 6, the expression of MCT1 looks higher in the epithelioid cell. It is better to indicate the positive staining by the arrow.
Author Response
Reviewer 1
Q1. The present study aimed to dissect in UM cell line (92.1) the role of lactate as either a metabolite or a signalling molecule using known modulators of HCAR1 and of lactate transporters. The authors found that lactate resulted in a significant decrease of cell proliferation and migration acting and switching cell metabolism towards oxidative phosphorylation. They showed the results were coupled with increased euchromatin content and quiescence in UM cells.
R1. We thank this reviewer for her/his time, courtesy and expert review of our manuscript.
Q2. In figure 3, the authors showed that lactate treatment induced a significant increase in mRNA expression and protein levels of MCT1 and HCAR1 and that these effects were reverted by AZD3965. Please explain how MCT1 the inhibitor block lactate induced the expression of MCT1. In Figures 3h and 3i, there is no big difference between the different groups.
R2. We believe that this reviewer is asking how lactate induces MCT1. We found that AZD3965 (an MCT1 inhibitor) reduces both HCAR1 and MCT1 expression levels (figure 3c-d). Moreover, we went deeper in this mechanism using agonist and antagonist of HCAR1 finding an HCAR1-dependent mechanism. We recognize that in representative blot in figure 3H a strong increase in HCAR1 can be observed in lactate exposed group whether no big differences can be observed between groups. We concluded that no significant changes are induced by lactate in cotreatment with 3-OBA and 3,5-DHBA. We apologize if we misinterpreted reviewer’s request, and we would like to emphasize here our availability to further fulfil reviewer’s concerns.
Q3. In figure 4, how about the expression of ND4, TFAM.
R3. We thank the reviewer for this consideration. We now proceeded to analyze the expression of ND4 and TFAM.
Q4. Why did the authors choose to detect the effect of lactate on the expression H3K18lac? And in figure 5, I suggest the authors detect the senescence markers in a different way to confirm the results.
R4. We thank the reviewer for this comment. The role of lactate as epigenetic player mediating chromatin relaxation was investigated by assaying H3K18lac accumulation. We propose this assay because of the availability of antibody given nowadays. Anti-H3K18lac is indeed the only one available on the market so far. Concerning the second point, we investigated the markers of quiescence and not senescence. It is worth noticing that these two processes are profoundly different (Terzi et al.; Mol Biol Rep 2016). For this purpose, as also discussed along the text, we had different hint prompting us to investigate if cellular quiescence markers were upregulated i) Cellular proliferation decrease; ii) increased mitochondrial activity; iii) Enlarged nuclei; iv) Increase in euchromatin rate. qPCR was used as a further way corroborating the overall picture depicted along the text.
Q5. In Figure 6, the expression of MCT1 looks higher in the epithelioid cell. It is better to indicate the positive staining by the arrow.
R5. We thank the reviewer for this advice. Firstly, we would like to apologize for the confusion generated. Indeed, we wrongly reported in the body text MCT1 and in figure’s caption MCT4. We now provide a revised version of the manuscript removing this error. In figure 6, the staining is MCT4 (top panel) and HCAR1 (bottom panel). Furthermore, in the epithelioid form the dark-brown spots correspond to pigment deposit of melanin, and it is not MCT4 positive staining (light-brown). We now pointed out this difference via arrowheads and arrows indicating positive staining.
Reviewer 2 Report
The title of the paper "Lactate rewrites the metabolic reprogramming of uveal melanoma cells and induces quiescence phenotype by lactylation" is much more appealing than the content. I'd say, that the title must be changed, because 1) most of the paper uses a single cell line as a model, but the authors expand it to melanoma cells in general and 2) the evidence for the role of lactilation is very poor.
Although the chapter 2.6 contains a figure (Fig. 6) and 5 lines of text describing the patients' data, corresponding to "Fifty-one primary UM samples ... collected ... from October 2009 to October 2019" (according to chapter 4.7), the figure 6 itself has no sign of such amount of samples. No data can be also found in supplementary too. Moreover, the statement "No written informed consent was necessary because of the retrospective nature of the study" provides ethical concerns. The only benefit of this chapter is formal ability to state, that the study was conducted not in a single cell line, however it shows just the data on expression of two proteins, and has little value for the mechanisms described.
As for the lactylation, it is truly supported by only one blot (Fig. 5D) and has no further information in Supplementary. Btw, the other supplementary blots have no indications of molecular weights and show only the blots corresponding to the paper, but the replicates should be provided too. Some of the blots are of rather bad quality, including dots on the relevant areas.
The lactylation itself is an interesting topic, but the authors didn't pay enough attention to it, which can be seen from the discussion.
"To dissect the mechanism by which lactate decreases cell proliferation we show here for the first time that its supplementation enhances H3K18 lactylation on UM cells, a phenomenon by which lactate promotes an euchromatin state, also characterized by enlarged nuclei. As per acetylation, we speculate that lactylation disrupt the electrostatic charge standing in between DNA phosphates and lysine, overall promoting chromatin relaxation [67]."
In fact, the paper [31], already elucidates the mechanism, which was further revealed in other papers including recent ones. So there is no need to speculate on a similarity with acetylation and cite the paper [67] here.
Importantly, the authors also evaluate only H3K18 lactylation but prefer discussing lactylation in general most of the times. One should use pan lactylation antibodies or mass-spec for such statements.
Finally, the statistics is described or used with little attention to the details, which also can be seen from figures. For example, the data representation in Fig 1C, 2C and similar graphs is good, since there are only 5 data points. Still not one-way, but two-way ANOVA should be used in the mentioned examples. In addition, t-test is a bad test for the statistically small numbers. Mann-Whitney U-test should be used instead. The details on statistics should be clearly stated in each case (in each graph too).
Clearly, the paper should be rewritten.
Author Response
Reviewer 2
Q1. The title of the paper "Lactate rewrites the metabolic reprogramming of uveal melanoma cells and induces quiescence phenotype by lactylation" is much more appealing than the content. I'd say, that the title must be changed, because 1) most of the paper uses a single cell line as a model, but the authors expand it to melanoma cells in general and 2) the evidence for the role of lactilation is very poor.
R1. We thank this reviewer for her/his time and effort in reviewing our manuscript. We agree with this reviewer that lactylation in the title may be misleading and we agreed to remove this term. We would keep uveal melanoma cells given that we used in vitro and patients’ tissue samples.
Q2. Although the chapter 2.6 contains a figure (Fig. 6) and 5 lines of text describing the patients' data, corresponding to "Fifty-one primary UM samples ... collected ... from October 2009 to October 2019" (according to chapter 4.7), the figure 6 itself has no sign of such amount of samples. No data can be also found in supplementary too. Moreover, the statement "No written informed consent was necessary because of the retrospective nature of the study" provides ethical concerns. The only benefit of this chapter is formal ability to state, that the study was conducted not in a single cell line, however it shows just the data on expression of two proteins, and has little value for the mechanisms described.
R2. We thank the reviewer for this critical comment. We now provide a revised version of the manuscript in which institutional review board statement was included and clarified and we also include the statement on informed consent. Please note that this section was wrongly stated due to a mistake by one of the authors. We also would like to underline at this stage that it is required for our Institutional review board to obtain informed consent from human subject to participate to any clinical study. We apologize for the inconvenient and we now provide a revised version of the manuscript that has been double checked for all these aspects.
We believe that figure 6 containing clinically relevant sections is corroborating our evidence. Given the revised version and the updated title, we believe this reviewer will found now the overall manuscript and statements appropriate.
Q3. As for the lactylation, it is truly supported by only one blot (Fig. 5D) and has no further information in Supplementary. Btw, the other supplementary blots have no indications of molecular weights and show only the blots corresponding to the paper, but the replicates should be provided too. Some of the blots are of rather bad quality, including dots on the relevant areas.
R3. We include in the manuscript and supplementary information compelling evidence of lactylation, although we agreed with reviewer’s suggestion (Q1-Q2) to temper our claims. Western blot is still a standard technique to demonstrate increased levels of specific proteins and we believe that blots are of publication quality. As per “instruction for authors” of this journal we provided with the first submission the uncropped, untouched, full original images of western blots.
Q4. The lactylation itself is an interesting topic, but the authors didn't pay enough attention to it, which can be seen from the discussion. "To dissect the mechanism by which lactate decreases cell proliferation we show here for the first time that its supplementation enhances H3K18 lactylation on UM cells, a phenomenon by which lactate promotes an euchromatin state, also characterized by enlarged nuclei. As per acetylation, we speculate that lactylation disrupt the electrostatic charge standing in between DNA phosphates and lysine, overall promoting chromatin relaxation [67]." In fact, the paper [31], already elucidates the mechanism, which was further revealed in other papers including recent ones. So there is no need to speculate on a similarity with acetylation and cite the paper [67] here. Importantly, the authors also evaluate only H3K18 lactylation but prefer discussing lactylation in general most of the times. One should use pan lactylation antibodies or mass-spec for such statements.
R4. We appreciate the reviewer’s comment on the hot-topic we are focussing in this manuscript. We now provide a revised version of the manuscript in which the title has been reviewed and the sentence cited by this reviewer has been modified accordingly and focussing on H3K18 lactylation. We also removed ref 67 as per reviewer’s suggestion.
Q5. Finally, the statistics is described or used with little attention to the details, which also can be seen from figures. For example, the data representation in Fig 1C, 2C and similar graphs is good, since there are only 5 data points. Still not one-way, but two-way ANOVA should be used in the mentioned examples. In addition, t-test is a bad test for the statistically small numbers. Mann-Whitney U-test should be used instead. The details on statistics should be clearly stated in each case (in each graph too).
R5. Please not that we are considering just treatment as variable. No statistical considerations are reported versus time (i.e. 24 h versus 48 h). Data are indeed % of wideness and are reported versus untreated control at the same timepoint and not versus control at a different timepoint. As such, the only variable in these cases is the variable “treatment” for n > 2 groups. One-way Anova is an appropriated statistical test to use with Holm-Sidak’s multiple comparisons test. Please also note that t-test for normal distribution is a suitable test for comparison of normal distributed value of n = 2 groups and we would prefer using this test.
Round 2
Reviewer 1 Report
N/A
Author Response
Thank you
Reviewer 2 Report
Although the authors made some valuable improvements during the revision, the current manuscript is still far from being acceptable.
First of all I have to insist on correction of the figure(s) at pages 10-11 of the PDF available to me as a reviewer. Currently it's practically impossible to evaluate these results and the corresponding text, as there likely happened a collision of the Fig 6 and Fig 5. It's also difficult to understand, where the fig 6 is cites, where is the Figs 5A-C are and why the lines 232-233 interfere between the figure (5) and its description.
However, the second point, to which I'd like to drive attention is the statistics. It's rather poorly described and the figures such as Fig. 1C, 2C, 2F, etc. imply questions on the analysis. A reader should be provided with the precise information on the statistics in the figure description. You may even divide graphs in the mentioned figures into two separate graphs for more clarity. The current representation imply using of the two-way ANOVA, brief evaluation of time factor and other data analysis. Importantly, the data of Fig. 2A, 2C, 2D and 2F must be analyzed with two-way ANOVA and the effect of lactate and the other used drugs should be assessed as factors and their interaction. These data may be important from the mechanistic point of view.
As I've written before, the t-test is not good to use with small samples, which is the case in this paper. The authors mention the normal distribution, however they didn't describe it in the "Methods" at all. I still suggest using U-test instead, because it provides more trustable statistical results for small samples.
Finally, I regret, that the Discussion lacks any schemes or figures, describing the obtained results. There might be a graphical abstract, but as a reviewer I strangely enough can't reach it. Anyway, all the obtained data can provide a good material for a schematic conclusion, supporting the results.
Despite made improvements, I have to reject the manuscript in its current form.
Author Response
Q1. Although the authors made some valuable improvements during the revision, the current manuscript is still far from being acceptable.
A: We appreciated his/her time in re-reviewing our manuscript. We think we now provided an updated version of the manuscript, now answering also to the second round of revision point by point.
Q2. First of all I have to insist on correction of the figure(s) at pages 10-11 of the PDF available to me as a reviewer. Currently it's practically impossible to evaluate these results and the corresponding text, as there likely happened a collision of the Fig 6 and Fig 5. It's also difficult to understand, where the fig 6 is cites, where is the Figs 5A-C are and why the lines 232-233 interfere between the figure (5) and its description.
A: We thank the reviewer for this comment. We now correct the file.
Q3. However, the second point, to which I'd like to drive attention is the statistics. It's rather poorly described and the figures such as Fig. 1C, 2C, 2F, etc. imply questions on the analysis. A reader should be provided with the precise information on the statistics in the figure description. You may even divide graphs in the mentioned figures into two separate graphs for more clarity. The current representation imply using of the two-way ANOVA, brief evaluation of time factor and other data analysis. Importantly, the data of Fig. 2A, 2C, 2D and 2F must be analyzed with two-way ANOVA and the effect of lactate and the other used drugs should be assessed as factors and their interaction. These data may be important from the mechanistic point of view.
A: We thank the reviewer for the comment. We agree with reviewer suggestion, and we divided now the figures providing a histogram per each time point, and described the statistics used also in figure legends
Q4. As I've written before, the t-test is not good to use with small samples, which is the case in this paper. The authors mention the normal distribution, however they didn't describe it in the "Methods" at all. I still suggest using U-test instead, because it provides more trustable statistical results for small samples.
A: We appreciated reviewer’s comment. In agreement with this we amended the statistical analysis. We performed indeed the U-test, providing a revised statistical analysis as suggested by the reviewer. We also performed the same analysis on all the data set, where required, finding no changes in significance compared to the old t-test.
Q5. Finally, I regret, that the Discussion lacks any schemes or figures, describing the obtained results. There might be a graphical abstract, but as a reviewer I strangely enough can't reach it. Anyway, all the obtained data can provide a good material for a schematic conclusion, supporting the results.
A: We appreciated the reviewer suggestion. In the updated revised manuscript, we now provide a fully comprehensive graphical abstract, where the main message of the paper has been illustrated.
Round 3
Reviewer 2 Report
Indeed, the manuscript was further improved, which is very encouraging. Some, likely final, modifications remaining from the previous step are needed. Without further/new comments I'd like to point the authors' attention to the suggested corrections, which were not fully responded.
Q3 previous.
However, the second point, to which I'd like to drive attention is the statistics. It's rather poorly described and the figures such as Fig. 1C, 2C, 2F, etc. imply questions on the analysis. A reader should be provided with the precise information on the statistics in the figure description. You may even divide graphs in the mentioned figures into two separate graphs for more clarity. The current representation imply using of the two-way ANOVA, brief evaluation of time factor and other data analysis. Importantly, the data of Fig. 2A, 2C, 2D and 2F must be analyzed with two-way ANOVA and the effect of lactate and the other used drugs should be assessed as factors and their interaction. These data may be important from the mechanistic point of view.
A: We thank the reviewer for the comment. We agree with reviewer suggestion, and we divided now the figures providing a histogram per each time point, and described the statistics used also in figure legends
Q3-new. Please also divide Fig. 4 B,D,F into two subgraphs as you did in 1C. No information on statistics is added in Fig. 4. No addition of two-way ANOVA factor analysis was made to Fig. 2A, 2C, 2D and 2F. No brief evaluation of time factors was incorporated.
Author Response
REVIEWER
Q1-Round 3
Indeed, the manuscript was further improved, which is very encouraging. Some, likely final, modifications remaining from the previous step are needed. Without further/new comments I'd like to point the authors' attention to the suggested corrections, which were not fully responded.
A1-Round 3
We sincerely thank this reviewer for her/his comments and careful revision of our manuscript. We are glad that she/he recognize the effort we placed in this manuscript and we believe that it is now significantly improved.
Q2-Round 3
Q3 previous. However, the second point, to which I'd like to drive attention is the statistics. It's rather poorly described and the figures such as Fig. 1C, 2C, 2F, etc. imply questions on the analysis. A reader should be provided with the precise information on the statistics in the figure description. You may even divide graphs in the mentioned figures into two separate graphs for more clarity. The current representation imply using of the two-way ANOVA, brief evaluation of time factor and other data analysis. Importantly, the data of Fig. 2A, 2C, 2D and 2F must be analyzed with two-way ANOVA and the effect of lactate and the other used drugs should be assessed as factors and their interaction. These data may be important from the mechanistic point of view.
A: We thank the reviewer for the comment. We agree with reviewer suggestion, and we divided now the figures providing a histogram per each time point, and described the statistics used also in figure legends
Q3-Round 2. Please also divide Fig. 4 B,D,F into two subgraphs as you did in 1C. No information on statistics is added in Fig. 4. No addition of two-way ANOVA factor analysis was made to Fig. 2A, 2C, 2D and 2F. No brief evaluation of time factors was incorporated.
A2-Round 3
We now realized the error in statistical analysis reported in the previous version of the manuscript. We now re-analysed our dataset reported in figure 2.
We now provide a revised figure 4 splitting graphs in B, D and F in two subgraphs. We also re-analysed data in figure 4F using two-way ANOVA.
OLD Figure 4 B,D,F NEW Figure 4 B,D,F
We also changed the representation of our data including the 2 variables (Lactate or AZD3965 / Lactate or 3-OBA) and we now report all comparisons that are statistically significant using two-way ANOVA and Holm Sidak posthoc test for multiple comparisons test.
Figure 2A
OLD (one-way ANOVA) NEW (two-way ANOVA)
Figure 2C
OLD (one-way ANOVA) NEW (two-way ANOVA)
Figure 2D
OLD (one-way ANOVA) NEW (two-way ANOVA)
Figure 2F
OLD (one-way ANOVA) NEW (two-way ANOVA)
Please note that we also re-analysed data included in figure 3 using two-way ANOVA.
